# On convolutional neural networks for selection inference: Revealing the effect of preprocessing on model learning and the capacity to discover novel patterns

**Ryan M. Cecil**[1,2]*, **Lauren A. Sugden**[1]

**1** Department of Mathematics and Computer Science, Duquesne University, Pittsburgh, Pennsylvania, United States of America, **2** Department of Statistics, University of Pittsburgh, Pittsburgh, Pennsylvania, United States of America

* RMC144@pitt.edu

## Abstract

A central challenge in population genetics is the detection of genomic footprints of selection. As machine learning tools including convolutional neural networks (CNNs) have become more sophisticated and applied more broadly, these provide a logical next step for increasing our power to learn and detect such patterns; indeed, CNNs trained on simulated genome sequences have recently been shown to be highly effective at this task. Unlike previous approaches, which rely upon human-crafted summary statistics, these methods are able to be applied directly to raw genomic data, allowing them to potentially learn new signatures that, if well-understood, could improve the current theory surrounding selective sweeps. Towards this end, we examine a representative CNN from the literature, paring it down to the minimal complexity needed to maintain comparable performance; this low-complexity CNN allows us to directly interpret the learned evolutionary signatures. We then validate these patterns in more complex models using metrics that evaluate feature importance. Our findings reveal that preprocessing steps, which determine how the population genetic data is presented to the model, play a central role in the learned prediction method. This results in models that mimic previously-defined summary statistics; in one case, the summary statistic itself achieves similarly high accuracy. For evolutionary processes that are less well understood than selective sweeps, we hope this provides an initial framework for using CNNs in ways that go beyond simply achieving high classification performance. Instead, we propose that CNNs might be useful as tools for learning novel patterns that can translate to easy-to-implement summary statistics available to a wider community of researchers.

## Author summary

The ever-increasing power and complexity of machine learning tools presents the scientific community with both unique opportunities and unique challenges. On the one hand, these data-driven approaches have led to state-of-the-art advances on a variety of research

**Data Availability Statement:** All code for reproducing the results in this paper is available at https://github.com/ryanmcecil/popgen_ml_sweep_detection.

**Funding:** This work was supported by the Duquesne University Faculty Development Fund (LAS) and the Wimmer Family Foundation (LAS). The funders had no role in software design, data collection or analysis, decision to publish or the preparation of the manuscript.

**Competing interests:** The authors have declared that no competing interests exist.

problems spanning many fields. On the other, these apparent performance improvements come at the cost of interpretability: it is difficult to know how a model makes its predictions. This is compounded by the computational sophistication of machine learning models which can lend an air of objectivity, often masking ways in which bias may be baked into the modeling decisions or the data itself. We present here a case study, examining these issues in the context of a central problem in population genetics: detecting patterns of selection from genome data. Through this application, we show how human decision-making can encourage the model to see what we want it to see in various ways. By understanding how these models work, and how they respond to the particular way in which data is presented, we have a chance of creating new frameworks that are capable of discovering novel patterns.

## Introduction

Over the last few decades, methods for detecting positive selection from present-day genome data have become more sophisticated and more powerful, yielding a large suite of tools designed for this particular evolutionary scenario. Early summary statistics were designed to detect the signature of strong selection, or a "selective sweep" based on its effect on the site frequency spectrum, relying on a signature of over-abundance of low- and high-frequency derived alleles [1, 2]. These were followed by tests capitalizing on the signature of elevated linkage disequilibrium across the site of a sweep, as calculated by allele frequency correlations [3, 4]. Later, the availability of phased genetic data paved the way for the broader use of statistics based on haplotype frequency [5, 6], and then for summary statistics designed to detect extended haplotype homozygosity (EHH) [7–11]. The 2010s saw the development of composite classification models designed to increase the power to detect sweeps by combining multiple summary statistics in various supervised classification frameworks, training these classifiers on simulated genomic data [12–17].

More recently, advances in machine learning have presented an exciting prospect: these models might be able to learn patterns of natural selection directly from raw data, thereby extracting more information than is available through current handcrafted summary statistics. To this end, multiple deep learning architectures have recently been proposed that can detect signatures of natural selection from sequence data with high accuracy [18, 19]. In addition, deep learning has been applied to other population genetics problems such as identifying recombination hotspots [20] and adaptive introgression [21], and distinguishing between soft sweeps and balancing selection [22]. Especially in these areas, where the well of theoretically-driven summary statistics is less deep than for selective sweeps, models that can learn new patterns and carry out these tasks without reliance on statistics provide an exciting way forward. Beyond achieving high performance at these tasks, we further propose that these models can help point us in useful directions: through advances in deep learning interpretation methods [23–25], we have the potential to conceive of new easy-to-implement summary statistics based on learned features.

While deep learning approaches have become state-of-the-art for a variety of tasks such as image processing [26–29], natural language processing [30–32], game playing [33, 34], and protein folding [35, 36], these models are often highly complex and not well understood. A few consequences of this are worth consideration. First, there are infinitely many possible model architectures, and identifying the optimal architecture for a given problem can be difficult. Second, once an architecture is chosen, it can be difficult to explain the resulting model's

predictions. Third, deliberate preprocessing decisions that determine how data is presented to the model can affect the decision rules learned by the model. On one hand these decisions often improve model performance by allowing the model to access known signals within the data. On the other hand, these decisions can limit the model to focusing only on a previously known signal, thereby constraining the model from learning something new.

The recently proposed state-of-the-art deep learning approaches for detecting selective sweeps are convolutional neural networks (CNNs) [37–40]. These approaches vary widely in their choice of architecture, including the number of layers, the number and size of convolution kernels, and methods of regularization (see S1 Table). While all methods report high accuracy on simulated testing data (typically generated in the same way as the training data), what is as yet unexplored is how these models work, what preprocessing decisions they might be sensitive to, and what insights they might be able to provide to our current theory. In the field of sweep detection, given the wide range of tools already available, and the well-studied theory developed over decades, one possibility is that the complexity of CNNs is unneccessary for improving performance. On the other hand, if they do improve upon our current suite of tools, then by understanding what additional information is deemed valuable by these complex models, it could be possible to improve the design of easy-to-use summary statistics beyond their current capabilities.

To address these questions, we first develop a very low-complexity CNN that we name "mini-CNN," which maintains high performance while allowing for more transparency into the classifier's inner workings. In addition, to interpret the more complex models, we draw on recent advances from the field of explainable AI [23–25], in particular using post-hoc interpretability methods built for visualizing and explaining feature importance for predictions after model training. Using these tools, we show empirically that typical preprocessing pipelines induce the CNN models for sweep detection to closely mimic handcrafted summary statistics; depending on how the data is presented to the models, their decision rules are particularly similar to Garud's H1, which typically performs at least as well at this classification task, or they pay closest attention to the number of segregating sites in a window, mimicking one of the earliest and simplest measures of genetic diversity, Watterson's theta.

We note, finally, that our analyses here are restricted only to the case of recent hard selective sweeps. CNNs have been shown to be effective across a wide range of applications even within evolutionary biology, including for detecting introgression [21], soft sweeps [41], and balancing selection [22]. We hope that this work can serve as a case study for how these other CNNs might be explored and mined for insights. We do not intend to generalize our particular results to those scenarios.

## Results

### CNNs for detecting selection

Multiple CNN methods have been proposed to solve tasks within the field of population genetic inference, with each following a common approach. First, training data is generated through a demographic model of choice using simulation software. Each sample, consisting of a set of haplotypes, is then organized into an image where rows represent individual haplotypes and columns represent genomic sites. These images undergo processing, and are standardized to a uniform width using varying techniques. Then, a CNN architecture is chosen and the model is trained and tested on the simulated genomic image data. Afterwards, the model may be applied to genome data from real populations. Fig 1A illustrates a general example of the process of applying a CNN to sampled haplotype data.

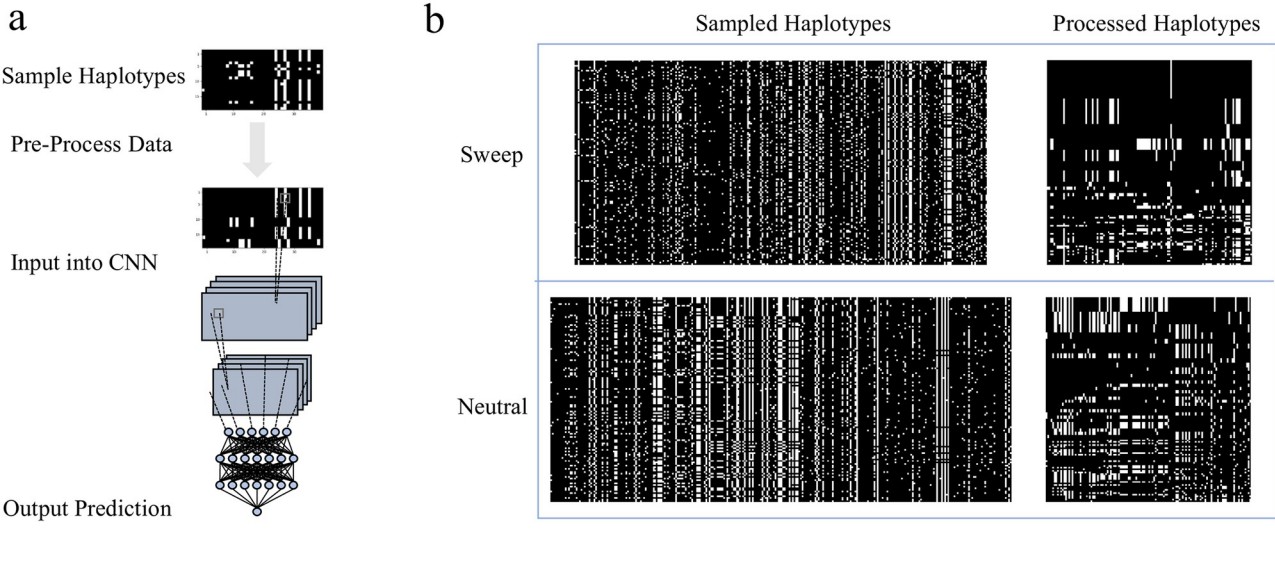

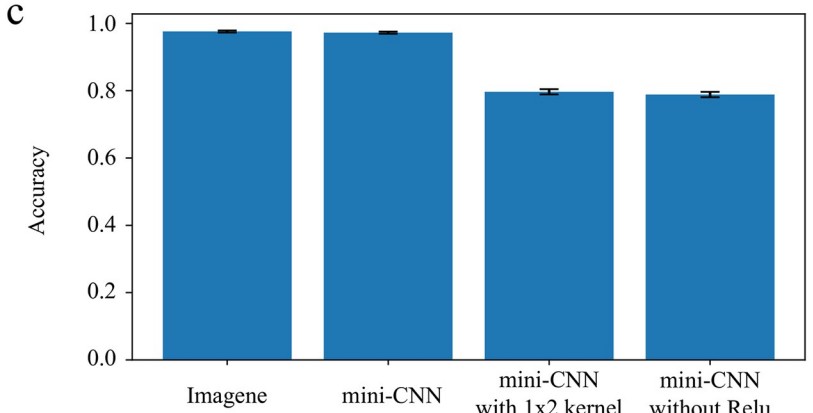

**Fig 1. Typical workflow for CNNs for sweep detection, and construction of a minimal CNN.** A: Raw sequence data is represented as an image with rows corresponding to sampled haplotypes and columns corresponding to segregating sites. Pre-processing steps ensure that images are of uniform size, as required for the CNN, and can also rearrange rows and columns to bring out useful features for the CNN. The CNN itself can comprise a number of convolution, activation, and pooling layers, followed by a number of fully connected "dense" layers, ultimately resulting in a single numerical output representing a prediction (neutral or sweep). B: Examples of images representing raw data (left) and pre-processed data (right). In this case, we follow the conventions of Torada *et al.* [38], where the images are converted to major/minor polarization and filtered by allele frequency, the rows are sorted by genetic similarity, bringing the most common haplotype to the top, then the image undergoes width standardization. C: Accuracy values on a balanced test set simulated from our single-population demography (error bars represent 95% confidence intervals). mini-CNN maintains comparable performance to Imagene despite a much simpler architecture with fewer layers and fewer, smaller kernels. mini-CNN employs a single 2x1 kernel; we find that swapping this out for a 1x2 kernel reduces performance significantly, as does reducing mini-CNN any further, for example by removing the Relu activation function. See Fig A and Table A in S1 Text for architecture details and intermediate architectures between Imagene and mini-CNN.

S1 Table contains an overview of recently proposed CNN methods, including details of their simulations, preprocessing pipelines, and network architectures. Within the preprocessing pipeline, there exist three main approaches to generating images of fixed width: using an image resizing algorithm, sampling haplotypes of fixed width (either by including invariant sites, or allowing for varying genomic ranges), and zero-padding (see Methods). Notably, either before or after this resizing occurs, almost all of the methods [37, 38, 40] sort the haplotypes of the image by genetic similarity to create visible block-like features within the genetic

image, with a block representing the most common haplotype at the top (see Fig 1B). In general, this has been shown to improve the performance of the model [37, 38]. Interestingly, the choice of CNN architecture varies widely across these works. The number of kernel dimensions, kernels, and convolutional layers alone range from 1D-2D, 4–256, and 1–4, respectively. Some models employ a branched architecture, allowing for additional inputs beyond the haplotype image [37]; since this presents an additional challenge for interpretation, we do not analyze branched architectures here. Together with varying architecture, models in the literature also vary in terms of the chosen demographic model and simulation software; altogether this makes direct comparison between the approaches tricky. A close look into how these CNN models make their predictions would aid in this regard.

To take a closer look into this class of CNN methods, we chose to implement and analyze the Imagene [38] model as a representative example, employing 2-dimensional convolutional kernels, and representing neither the least nor the most complex architecture. Like the other approaches, it sorts rows based on genetic similarity during preprocessing.

## Construction of mini-CNN and interpretation of its output

Interpretation of CNNs with complex architectures is notoriously difficult, so we first attempted to construct a minimal CNN that maintained high performance at the task of distinguishing between sweep and neutral simulations. Beginning with the Imagene model [38] as a representative example of CNNs for sweep detection, Table A in S1 Text shows the accuracy values as we reduce and remove the convolution, dense, ReLU, and max pooling layers for simulations of a single-population demographic model with a selection coefficient of $s = 0.01$. We find that the model with a single 2x1 kernel, followed by ReLU activation, then a single unit dense and sigmoid activation layer, maintains high performance (a difference of 0.3% accuracy to Imagene). We call this model mini-CNN (see Fig 1C for performance, and Fig A in S1 Text for visualization of this model). Further simplification, such as removing the ReLU activation, results in a 18.4% accuracy loss.

Because of its simplicity, mini-CNN lends itself more easily to visual interpretation. In Fig 2A, we show the kernel weights for the 2x1 kernel, along with the dense weights map, which helps visualize the importance of different regions of the input image. The 2x1 kernel detects differences between two consecutive rows at a given locus, while the dark band at the top of the dense weights map, representing negative weights, indicates that the model interprets row-to-row differences near the top of the image as evidence against a sweep. Fig 2B illustrates how the model acts on an example sweep image. Because of row-sorting by genetic similarity, the large haplotype block at the top of the image does not contain row-to-row differences, leading to many 0's (black pixels) at the top of the image after the 2x1 kernel convolution and ReLU activation. While 1's (white pixels) in these positions would provide evidence for neutrality, these 0's point the model in the opposite direction after the dense weights are applied.

Examining whether this pattern holds with more complex models requires alternative approaches; to this end, we applied an explanatory tool called SHAP [42] to the full Imagene model. SHAP values are a measure of the change in model prediction conditioned on the particular features, allowing us to see which pixels are most influential. Fig 2C shows SHAP values for each pixel within two example training images—one neutral, and one sweep—as well as what these SHAP values look like on average across 1000 neutral and 1000 sweep images. We see a similar pattern here, in which the regions that contribute the most to the model prediction appear near the top of the image, coinciding with the largest haplotype blocks.

As a control, we include SHAP diagrams for Imagene trained on unsorted data in Fig B in S1 Text; when the row-sorting is not included as a preprocessing step, this pattern disappears

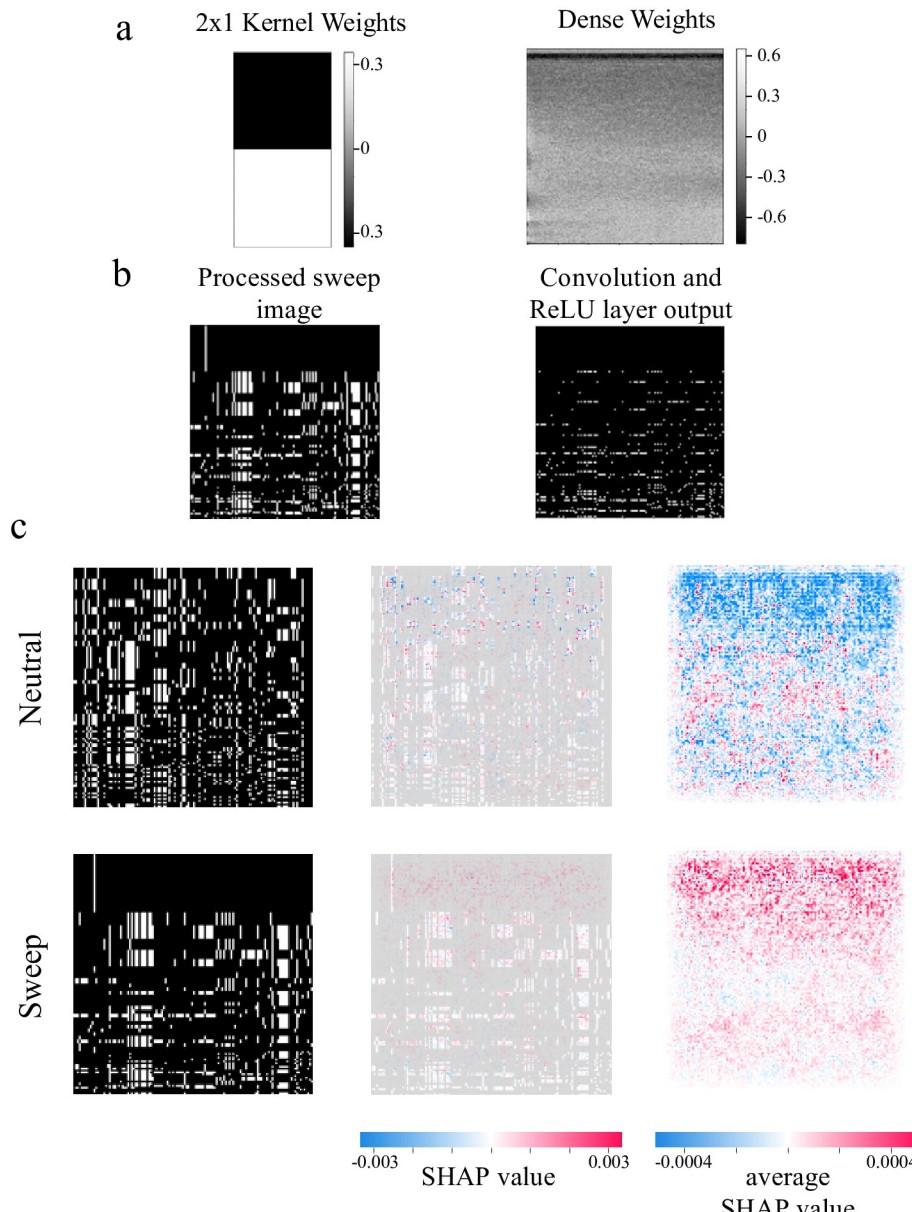

**Fig 2. Opening the black box: Visual explanations of mini-CNN and Imagene.** A: Trained parameter values for mini-CNN. The 2x1 kernel detects differences between consecutive rows. The dense weights map illustrates the linear weights that are applied to the output of the convolution and ReLU layer (depicted in panel B). The black band at the top of the weights indicates that the model is more likely to predict the image as neutral if there exists variation among the top rows. B: Example of a pre-processed image and its corresponding output after the convolution layer and ReLU activation. C: Visualization of Imagene with SHAP explanations. From left to right are examples of neutral and sweep processed images, SHAP values for the two image examples, and average SHAP values across 1000 neutral and sweep images. A negative SHAP value (blue) indicates that the pixel of interest contributes toward a prediction of neutral, while a positive SHAP value (red) indicates that the pixel of interest contributes toward a prediction of sweep. Similar to the black bar in the dense weights map in panel a, the large SHAP values located in the top region of the average Shap images indicate that Imagene focuses on the top block of the image to make its prediction.

and it becomes difficult to discern any relevant patterns. It is notable that despite this, Imagene still performs better than chance, with an accuracy of around 75%. A possible explanation for this is that there is still an elevated chance of row-to-row similarity after a sweep, even if these similarities are not deliberately gathered at the top of the image. Interestingly, mini-CNN does not perform as well as Imagene on unsorted data; in particular, while the number of layers can be reduced, we find that four 3x3 kernels are necessary to maintain the performance of Imagene (see Table B in S1 Text). Further analysis of this result will certainly benefit from advances in interpretive and explanatory tools that can provide insight into more complex models.

## Comparison to summary statistics

The pattern we observe in which the CNN models pay the most attention to the rows at the top of the row-sorted image is not surprising; this coincides with existing theory about the signature left behind by a selective sweep; in particular, a sweep induces long shared haplotype blocks [7]. This is the basis of many current hand-crafted summary statistics, including iHS [8], Garud's H statistics [9], nSL [43], and XP-EHH [10], with the last statistic requiring a reference population.

In Fig 3A, we measured the Spearman rank correlation between Imagene and mini-CNN output, and that of two commonly-used haplotype summary statistics, iHS and Garud's H1 (see also Fig C in S1 Text). We found that Garud's H1 in particular was highly correlated with both Imagene and mini-CNN. We also examined the correlation between the binary classification of each pair of methods (i.e. the proportion of simulations on which the classifiers agree; see Fig 3B). Here we find that Garud's H1 agrees with both Imagene and mini-CNN 97% of the time in the single-population demographic model, and 89% of the time in the three-population demographic model.

Fig 3C illustrates the design of Garud's H1, alongside a training image pre-processed with genetic-similarity-based row sorting common to CNNs for sweep detection. Garud's H1 operates by identifying shared haplotype blocks in a region, and computing the sum of squared haplotype frequencies. In the context of a pre-processed image then, Garud's H1 is sensitive to the number of identical rows present at the top of the image, raising the possibility that in the process of row-sorting, the CNN is being set up to mimic the behavior of Garud's H1.

In Table 1, we compare the accuracy of the CNN approaches and Garud's H1 for the single-population demographic model. For two selection coefficients representing low and moderate selection strength, s = 0.005 and s = 0.01, Garud's H1 actually *outperforms* both Imagene and mini-CNN, a pattern that is maintained even when these models are trained on images with a larger number of haplotypes (see Table D in S1 Text).

We note here that in evaluating the performance of Garud's H1, along with other window-based summary statistics, it is important to consider the window size as it relates to the expected sweep footprint. Following Harris *et al.* [44], and using our single-population demographic model parameters, we estimate a sweep footprint on the order of 50kb for *s* = 0.01. Our simulations, which span 80kb, should therefore represent a realm that allows for good, if not optimal, summary statistic performance. With a much larger window size, their performance might be expected to decline, while we may expect the CNNs to maintain their performance and learn to ignore columns outside of the sweep footprint.

## Performance in simulated human populations

A potential explanation for the performance of Garud's H1 relative to the CNNs, and the performance of mini-CNN relative to Imagene, is that the single-population demographic model simulated with `msms` results in sequence data that is less noisy than genuine human genome

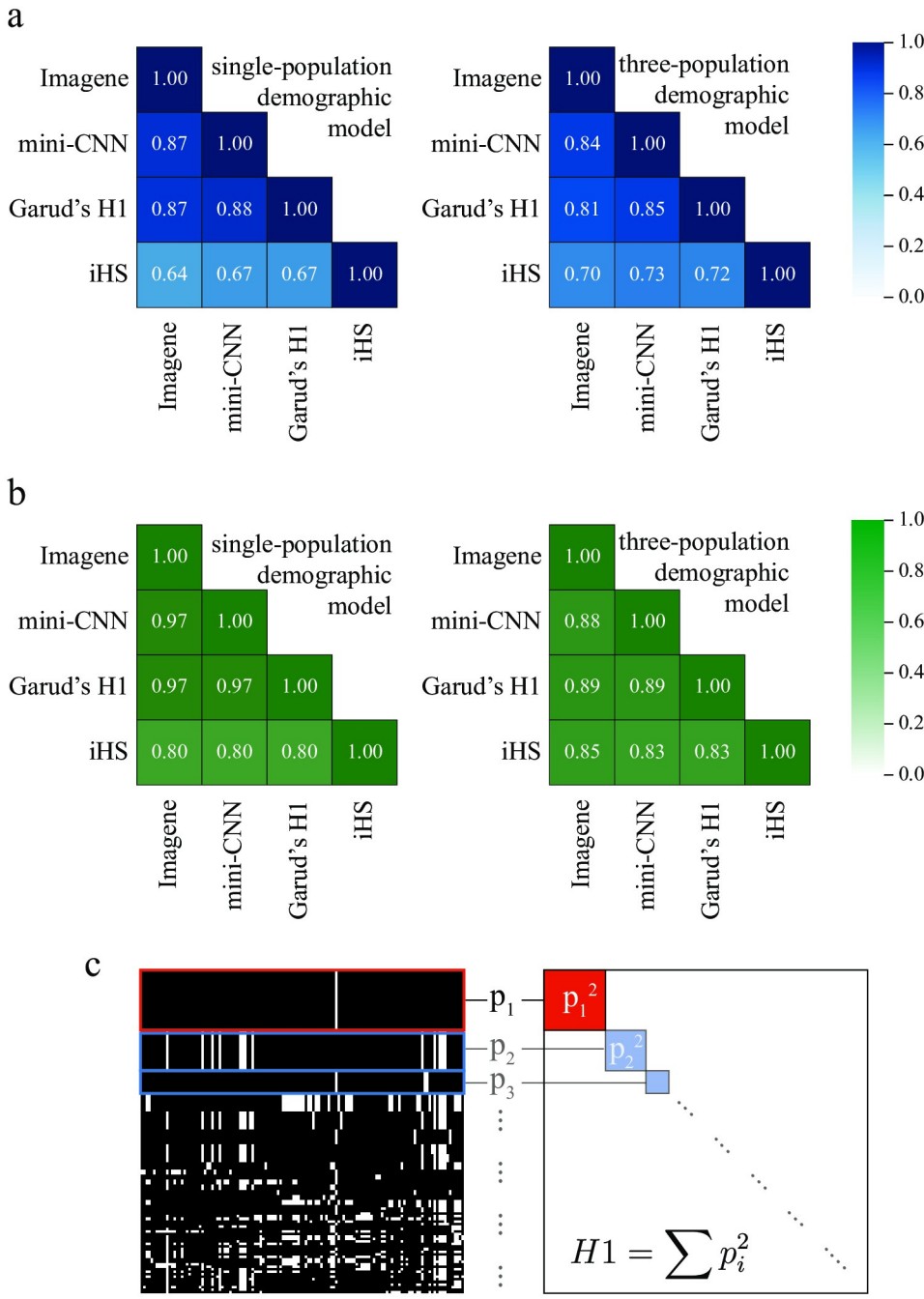

**Fig 3. Comparison of CNNs with Garud's H1 and iHS.** A: Spearman rank correlation matrices comparing continuous output of CNN approaches and two summary statistics (Garud's H1 and iHS) that measure haplotype homozygosity. Left: single-population demographic model simulated with msms. Right: three-population demographic model [45] with sweep in CEU, simulated with SLiM. B: Correlation matrices comparing binary output of CNN appraches and two summary statistics. Left: single-population demographic model simulated with msms. Right: three-population demographic model [45] with sweep in CEU, simulated with SLiM. Numbers shown are the proportion of simulations for which the two methods return the same classification (sweep or neutral), using optimal thresholds based on a training set. C: An example of a pre-processed image with row-sorting by genetic similarity (left). The haplotype with the highest frequency is brought to the top of the image. Calling this haplotype frequency $p_1$, and the remaining haplotype frequencies $p_2, p_3, \ldots$, Garud's H1 statistic computes the sum of squared haplotype frequencies, $\sum p_i^2$. Both the CNNs and Garud's H1 pick up on the signature of a high-frequency haplotype.

**Table 1. Performance comparison.** Performance values of multiple methods that were trained and tested on the same demographic model and selection coefficient. The performance values were found by computing the accuracy of the trained model on a balanced, held-out set. See Table C in S1 Text for additional models and summary statistics. These accuracies are also visualized in Figs D and E in S1 Text. Binomial-derived standard errors for these proportions are at most 0.5% for the single-population demographic model, and 1.1% for the three-population demographic model.

| | Single population demographic model | | Three-population demographic model, sweep in YRI | | Three-population demographic model, sweep in CEU | |
|---|---|---|---|---|---|---|
| Selection coefficient (s) | 0.01 | 0.005 | 0.01 | 0.005 | 0.01 | 0.005 |
| Imagene | 97.60 | 56.00 | 82.30 | 55.60 | 84.80 | 53.00 |
| Mini-CNN | 97.30 | 58.10 | 80.50 | 51.40 | 84.30 | 55.30 |
| Garud's H1 | **98.15** | **61.07** | **84.70** | 54.25 | **89.15** | 55.65 |
| DeepSet | 75.67 | 50.96 | 74.40 | **61.35** | 75.25 | **57.35** |

data. This could produce selective sweep signals that are in effect too easy to detect, allowing for less complex models to perform on par with those that are more complex. Indeed, published CNNs for sweep detection (S1 Table) have trained and tested their models on a variety of simulated demographic models, allowing for the possibility that these more complex models are necessary in more realistic human contexts.

To test these models on more realistic data, we used simulation software SLiM to simulate data under the three-population demographic model of Gravel *et al.* [45] which models the joint history of 1000 Genomes YRI, CEU, and combined CHB+JPT populations. This demographic model includes population bottlenecks and exponential growth, as well as migration among all three populations in the last 50,000 years. In Fig F in S1 Text, we show that the simulated neutral data replicate the site frequency spectrum of real 1000 Genome data [46] much more closely than the neutral simulations from the single-population model. We also simulate two selection coefficients as above: $s = 0.01$ and $s = 0.005$, to investigate the compounded effect of more subtle sweeps. As is expected due to the more complex demography, the performance of all three methods declines relative to their performance on the single-population demographic model. For stronger sweeps ($s = 0.01$), mini-CNN maintains a close performance to Imagene (a difference of 1.8% accuracy in the worse case), and for weaker sweeps ($s = 0.005$), mini-CNN's performance is 4.2% lower than that of Imagene when the sweep is in YRI, and 2.3% higher when the sweep is in CEU (see Table 1).

Notably, the patterns we observed previously regarding Garud's H1 hold in this context as well. Garud's H1 outperforms both Imagene and mini-CNN on stronger sweeps, and performs on par with both on weaker sweeps. We note that all three methods seems to be near the edge of their detection range for these weaker sweeps (accuracy approaching 50%). In Fig 3A, we see that similar to the single-population demography case, Garud's H1 is highly correlated with both Imagene and mini-CNN, and in Figs G and H in S1 Text, we show that the dense weights map (in the case of mini-CNN) and SHAP values (in the case of Imagene) display the same trends that are shown in Fig 2: the models are most attentive to the haplotype blocks near the top of the image.

## Effects of image-width standardization on CNN learning strategies

Following Torada *et al.* [38], we used an image resizing algorithm to attain genetic images of fixed width, but other methods have been proposed for this (see S1 Table). To test whether the relationship between Garud's H1 and the CNN models holds across other standardization methods, we implemented two other strategies: zero-padding, similar to the approach used in Flagel *et al.* [37], and trimming images down to a common width.

In Fig 4A, we show examples of zero-padded sweep and neutral images, which display a common trend: because sweeps characteristically result in reduced haplotype diversity, an unprocessed sweep image typically contains fewer columns than an unprocessed neutral image, thus requiring more padding to attain a particular fixed width. Fig 4B shows the output of the convolution and ReLU activation layers, which retain the original zero padding of the

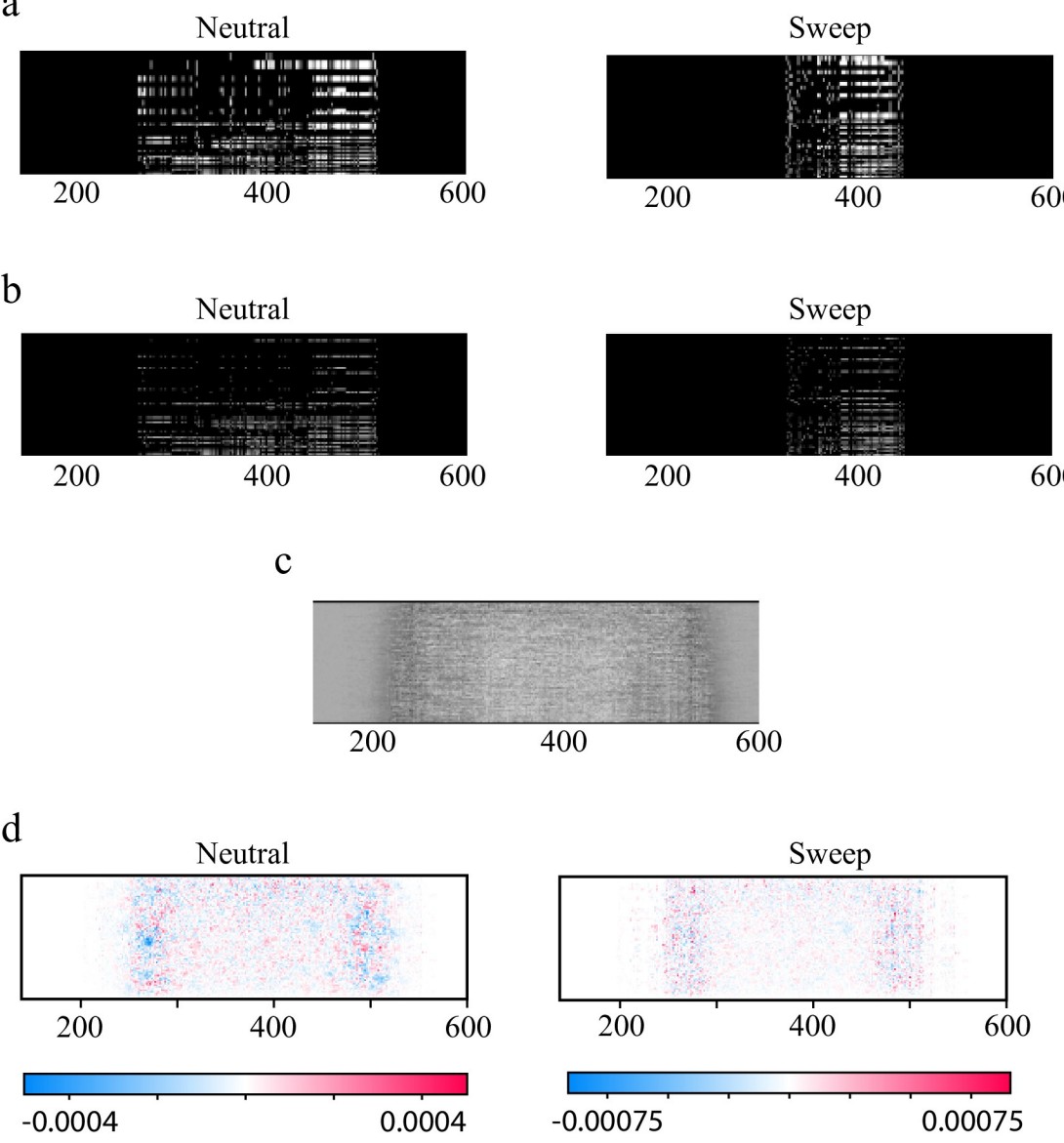

**Fig 4. Zero-padding induces the model to attend to the width of the unprocessed image.** A: Images processed with zero-padding from the three-population demographic model with a sweep in CEU with s = 0.005. A signature of a selective sweep is a reduction in heterozygosity across the population; this manifests as fewer columns in the raw image after filtering for variable sites. Padding the image with columns of zeros thus results in larger blocks on either side of a sweep image relative to a neutral image. B: Output of the convolution and ReLU activation of mini-CNN for the images in panel A showing that the large blocks of zeros on either side of the image persist. C: Trained dense weights map for mini-CNN for the simulations in panel a show that the model is attentive to the width of the padding. D: SHAP explanations for Imagene on the same simulations likewise show greater importance along the side edges of the pre-processed image. Note that all images have been cropped for ease of visualization; in all cases, columns outside of the cropped region contain no information. Uncropped images can be found in Fig M in S1 Text.

input. In Table C in S1 Text, across the various simulation types, the accuracy values for the CNN models paired with image-width standardization via zero-padding are reported. Notably, when zero padding is applied, the performance of the CNN methods surpass that of the Garud's H1, with the greatest improvements occurring for subtler sweeps. Further analysis suggests that this method of padding steers the model towards learning a signature very similar to Watterson's $\theta$, or more directly, the number of segregating sites $S$.

The effects of zero-padding are shown in Fig 4C and 4D, which contain visualizations for mini-CNN and Imagene models trained with zero-padding on the three-population demographic model with a sweep in YRI (see also Figs I and J in S1 Text). Interestingly, in Fig 4C, we see that in contrast to the dense weights map from Fig 2A, the negative weights (darker regions) lie along the columns of the map at two locations on the left and right. This indicates that mini-CNN is searching for variation at these locations as evidence against a sweep. Since there exists no variation within the zero padded columns, mini-CNN is likely to predict a sweep when an image contains a large amount of zero padding, which in turn is dependent upon the number of segregating sites in the unprocessed genetic image. Similar trends can be seen in the Imagene model with zero-padding. In Fig 4D, we show average SHAP values across 1000 neutral and 1000 sweep simulations for the Imagene model. Similar to above, the regions that contribute most to the model prediction lie along the columns of the left and right side.

In order to further elucidate this pattern, we implemented a summary statistic $S$ representing the number of segregating sites in the image. This statistic shares moderate-to-high correlation with the CNN models (Fig K in S1 Text), and has quite high accuracy, even surpassing the performance of the CNNs for some subtler sweep scenarios (Table C in S1 Text). We note here that this performance is likely inflated relative to what we would expect in real data. This is because our simulated data uses a fixed mutation rate, and $S$, as a sufficient statistic for the population scaled mutation rate [47] will be extremely sensitive to this parameter by design. In genuine genomes, by contrast, we see a vast range of mutation rates as a function of sequence context and other genome features [48]. Since the CNNs working with zero-padded data appear to be extracting a similar signature, we would also urge caution in interpreting the high accuracies of those models as well. We recommend that a CNN pipeline using zero-padding in preprocessing should be careful to include variable mutation rate parameters in the simulated training data, as is done in Flagel *et al.* [37] to limit overfitting.

We also implemented a trimming strategy (see Materials and methods), with results shown in Table C in S1 Text. With this image-width standardization strategy, we see similar results as with image resizing. In particular, mini-CNN performs on par with Imagene, with Garud's H continuing to outperform both. This method of standardization again makes the row-sorting the most salient signal, as interpreted by SHAP diagrams (see Fig L in S1 Text). We do see a decrease in performance relative to image resizing, which we attribute to the fact that we are trimming out potentially useful columns from some simulations. We note that this could likely be ameliorated by starting with even larger simulations to ensure that the minimum number of SNPs per simulation is above a desired threshold.

## Comparison to a permutation invariant network

The learned prediction methods of CNNs built for the detection of selective sweeps appear to critically depend on human-designed methods of preprocessing. These methods may include removing invariant columns, removing columns based on low minor allele frequency, and some method of image-width standardization as described in the previous section. Here, however, we focus specifically on the row-sorting procedures common to these CNNs (e.g. sorting by haplotype similarity). A chosen sorting procedure has the potential to not only influence

the learned decision rules, but also to restrain the power of the model to a level similar to that of previously-defined methods. If our goal is to use machine learning to build prediction methods with capabilities beyond that of current statistical approaches, then we must investigate ways to remove this human dependence.

One potential solution is to avoid the sorting step altogether, and instead implement a new machine learning architecture that is "permutation invariant," or agnostic to the ordering of the haplotypes. To investigate the potential of this approach, we implemented a permutation-invariant CNN architecture, similar to one used recently for recombination hotspot detection [20]. Following along with previous works on permutation-invariant architectures [49], we label this model "DeepSet". In Table 1 and Table C in S1 Text we compare the performance of the DeepSet model to the other CNNs and summary statistics. Notably, although the model underperforms in comparison to Garud's H1 on stronger sweeps, it outperforms the other methods when applied to subtler sweeps in the simulated human populations.

While we note that DeepSet has the best performance relative to other models on very subtle sweeps ($s = 0.005$ in the 3-population demographic model), performance is relatively low across all models and statistics in these scenarios. When we standardize images with zero-padding, DeepSet performance rises along with that of other CNNs (see Table C in S1 Text).

## Discussion

In this study, we examine the effects of different training methods, CNN architectures, and simulation settings on model performance at the task of detecting recent selective sweeps from genome data. We then attempt to explain these models in an effort to understand how they work and how they respond to various preprocessing strategies. An empirical analysis of the Imagene and mini-CNN models reveals that current architectures may ultimately rely on decision rules similar to that of Garud's H1 summary statistic when row sorting is present as a preprocessing step. This is supported by our mini-CNN architecture, which only computes differences across the rows. In addition, to improve performance of their CNN, Deelder *et al.* [40] optimized their architecture over various hyper-parameters, finding that the optimal model contained only one convolutional layer with four filters and two dense layers. Like mini-CNN, the complexity of their architecture is strikingly small in comparison to the other works that do not discuss architecture optimization.

In this work, we observe that low complexity models (including summary statistics) seem to perform on par with the more complex models for the task of detecting recent selective sweeps; this suggests the possibility that the additional complexity is unnecessary in this particular context. This would be very convenient; since summary statistics like Garud's H1 are simple to compute compared to CNN models, which require very large training sets as well as programming and machine learning expertise, this would put the power to detect sweeps into the hands of a broad swath of researchers. On the other hand, all of the models we implemented lose significant performance when selection coefficients become weaker, especially in the context of more complex demographic backgrounds. A similar pattern might be expected with older sweeps, or selection over a much longer timescale, which might necessitate different summary statistics, and would require different training simulations for the CNNs. Another example in this vein is distinguishing between soft and hard sweeps: a few preliminary experiments training Imagene and mini-CNN for this task resulted in fairly low accuracy (60.7% and 59.7% respectively), but did provide a couple of intriguing explanatory visualizations that may point in potentially fruitful directions for future research (Figs N and O in S1 Text).

This raises the possibility that more complex, tailored models *could* detect signatures that are as-yet unobserved, given the right architecture and simulation scheme. In particular, we

note that the supervised nature of CNNs allows for these models to be trained on complex demographies and evolutionary scenarios to approximate real-world data and draw out nuanced signals in ways that summary statistics may not be able to. To this point, CNNs have been shown to be effective in other population genetics applications including identifying recombination hotspots [20] and adaptive introgression [21], and distinguishing between soft sweeps and balancing selection [22], none of which we explore here. We also note that there are more complex architectures that exist in the literature, including a branched architecture that takes in positional data in addition to the haplotype image and employs kernels that span all haplotypes at once [37], potentially allowing for the model to learn more nuanced signatures. This additional model complexity makes interpretation more difficult, so we have not attempted it here. In all of these cases, the additional complexity of the CNNs may well be beneficial, especially in areas with less robust suites of summary statistics than the hard sweep case. Ideally, by examining these CNNs, insight could be gained that might result in more easily-applied tools in order to make them more broadly available; we hope that this study provides some inspiration for such investigations, which can dovetail with cutting-edge interpretive and explanatory tools as they emerge.

While the particular architecture of a CNN is an important factor in its performance, another aspect of the training and testing pipeline is also crucial: the preprocessing steps used to wrangle the data into a suitable set of input images. These are necessary in part because of the peculiarities of genomic data. To represent a collection of haplotypes as an image, there are multiple decisions to be made. These include, but are not limited to: whether to use major/minor or ancestral/derived polarization for alleles at a particular site, whether to include all genome sites, or only segregating sites, whether to put a threshold on minor allele frequency for inclusion as a site, how to deal with tri-allelic sites, and the order in which to present the haplotypes. Beyond that, for a CNN classifier, all training and testing images must have uniform width, while the number of genomic sites in a simulation may be variable, depending on the choices made above.

We find that the image-width standardization preprocessing step in particular has the potential to sway the learning strategy of the CNN. We see that standardizing via zero-padding results in a feature that is very salient to the CNNs: the wider blocks of zeros required to pad sweep images relative to neutral images. This is particularly extreme in our simulations, which do not have a variable mutation rate in contrast to the application in Flagel *et al.* [37]. Another strategy for image-width standardization, trimming images to match the minimum observed width, results in models that behave similarly to those trained on resized images. Particular care must be taken in this case to simulate large enough genome regions to ensure sufficiently many columns are left in the images.

We are particularly interested here in the practice of row sorting, common among CNN pipelines for sweep detection, in which high-frequency haplotypes are gathered together near the top of the image. We show this preprocessing step to be a huge factor in overall performance, a finding that is consistent with Torada *et al.* [38], who find a large jump in accuracy from unsorted to row-sorted images. Because the motivation for row-sorting is based on a signature of selection that is fairly well-understood (and serves as the basis for existing summary statistics), this raises the question of whether we are training complex models to see what we want them to see, a notion also raised by Flagel *et al.* [37]. Without this nudge, what other patterns *might* a CNN be able to find that would add to our theoretical understanding of natural selection?

There are two possible ways to address row-sorting as pushing the model in a specific direction: one can either learn an optimal sort as part of the model, or one can train a row-permutation invariant network. An immediate difficulty with the first approach is that permutations

are discrete and not amenable to gradient-based training algorithms which require model operations to be continuous and differentiable. Although it has been shown that is is possible to learn a sort by way of permutation matrices [50], it is not guaranteed to be feasible or effective in this context. A distinct approach is to instead remove the sorting step entirely and focus on permutation-invariant neural networks, commonly known as deepsets [49]. By applying permutation-invariant operations on the set of individual haplotypes, these methods are entirely agnostic to the order in which the haplotypes are presented. As evidence for the potential of this approach, Chan *et al.* [20] applied a row-permutation invariant network to the task of detecting recombination rate hotspots, finding an improvement in both training speed and classification performance over a standard CNN. Furthermore, as shown by Table 1, a simple permutation invariant network outperforms the Garud's H1 test statistic on realistic human data in the case of weak selection signals. These results provide hope for this method, and advances in architecture design and insights from other deep learning methods, such as the transformer [51], may lead to performance improvements.

Through an understanding of how high-performing machine-learning models learn patterns of evolution, we have the potential to extract new insights about these processes and the way our genomes are shaped by them. Ideally, these insights could suggest new frameworks with which to build summary statistics that can be easily constructed and deployed across a wide range of datasets. We note that this is a different perspective from which machine learning tools like CNNs are typically thought of; rather than existing as sophisticated tools only usable for completing specific tasks, we can treat them as tools that can help us learn *how* best to complete that task without the use of black-box models. We are not unique in this perspective; for instance, advances in reinforcement learning have recently lead to the development of less complex matrix multiplication algorithms discovered through the use of machine-learning models [52]. As theory concerning interpretability and explainability of machine learning improves, we can look forward to more opportunities for deriving insight into population genetics processes.

## Materials and methods

### Simulation of training and testing data

**Simulation of a single-population demographic model.**   Following the "three epoch" demographic model used in Torada *et al.* [38], we generated simulations of 128 chromosomes from a single population for an 80kb region with two instantaneous population size changes: starting with an ancestral population size of 10,000, the effective population size decreases to 2,000 at 3500 generations from present, then expands to 20,000 at 3,000 generations from present. We used a mutation rate of $1.5 \times 10^{-8}$, and a recombination rate of $1 \times 10^{-8}$. For sweep simulations, we used selection coefficients corresponding to $s = 0.01$ for heterozygotes (0.02 for homozygotes) and $s = 0.005$ for heterozygotes (0.01 for homozygotes), with the sweep beginning 600 generations ago, and the beneficial allele located in the center of the simulated region. Simulations were generated with msms [53]. We hereafter refer to this as the "single-population demographic model."

We also performed an experiment simulating 1000 chromosomes from a single population following the same demographic model to validate the patterns found in this manuscript. Original images were resized to 200x200 due to GPU memory constraints. Accuracy results for models trained on this dataset can be found in Table D in S1 Text; in short, the results mirror our results for the original 128-chromosome dataset.

**Simulation of 1000 Genomes populations.**   To simulate realistic human populations, we simulated 64 diploid individuals each from three populations representing 1000 Genomes [46]

YRI, CEU, and CHB populations, across 80kb regions. Our simulations used population sizes, split times, growth rate, and migration rate parameters inferred by Gravel *et al.* [45], based on 1000 Genomes exon and low-coverage data. We used a mutation rate of $2.363 \times 10^{-8}$ and recombination rate $1 \times 10^{-8}$, and assumed 25 years/generation. For sweep simulations, we used a dominance coefficient of 0.5, and selection coefficients corresponding to $s = 0.01$ for heterozygotes (0.02 for homozygotes) and $s = 0.005$ for heterozygotes (0.01 for homozygotes). The beneficial allele arises 600 generations ago in the center of the 80kb region. Simulations were generated with SLiM [54]. We hereafter refer to this as the "three-population demographic model."

## Preprocessing of haplotype data

For the purposes of training and testing a CNN, each set of simulated haplotypes had to be reshaped into a haplotype matrix (i.e. an image) of fixed width and length where the rows correspond to individual haplotypes and columns correspond to genome sites. Similar to the work in [38], we pre-processed the 128 sampled haplotypes from each simulation to matrix form by converting them to a binary image using major/minor polarization, removing loci with allele frequency less than 1%, standardizing the width of all images, and grouping the rows of the image into common haplotype blocks and then sorting the blocks based on haplotype frequency. To standardize the width across images, we tested three methods. First, following [38], we used the resizing algorithm from the `skimage` [55] python package to force a fixed width of 128 for each image ("image resizing"). The algorithm first applies a Gaussian filter to smooth the blocks of the image and avoid aliasing artifacts. Then, it performs a spline interpolation of order 1 for downsampling with 'reflect' padding applied to the boundaries. Second, following [37], we added columns of 0s to the edges to pad the images to a specified width ("zero-padding"). In this case, for each dataset, the chosen fixed width was defined by the maximum image width rounded up to the nearest multiple of 10. Third, we trimmed images to a fixed width ("trimming"). In this case, for each dataset, the chosen fixed width was defined by the minimum image width rounded down to the nearest multiple of 10.

## Implementation of deep learning models

The deep learning models were implemented using `Keras` [56] with `TensorFlow` [57] as the backend. Each CNN model implemented in this work is composed of a series of convolution, ReLU activation, and max pooling layers, a flattening layer, a series of dense and ReLU activation layers, and a final dense layer with a sigmoid activation function. At the start of training, the convolution and dense weights were initialized using random draws from a standard normal distribution and glorot uniform distribution [58], respectively. During training, L1 and L2 regularization terms each with a regularization penalty value of 0.005 were incorporated into the loss.

To briefly explore the potential of other deep learning architectures we also implemented a permutation invariant Deepset model [49]. The model contains two sets of convolutional and ReLU activation layers with 64 kernels of size 1x5, an averaging operation across the haplotypes, a flattening operation, 2 dense and ReLU activation layers with 64 units each, and a final dense layer with a sigmoid activation function. At the start of training, the convolution kernel and dense layer weights were initialized from the glorot uniform distribution.

For the purposes of training and testing, we partitioned each simulated dataset into a train (80%), validation (10%), and test set (10%), each with a balanced number of neutral and sweep images. Unless otherwise specified, there were 100,000 total samples for the single-population model (generated with `msms`), and 20,000 total samples in the three-population model

(generated with SLiM) (see Fig P in S1 Text for sample complexity analysis). The ML models were trained to distinguish between selective sweep and neutral images by minimizing the standard binary cross-entropy loss function. The Adam optimizer [59] with a mini batch size of 64, settings $\beta_1 = 0.9$ and $\beta_2 = 0.999$, and learning rate of 0.001 was used. All models were trained for 2 epochs where 1 epoch is a single pass through the training dataset. Each model was trained 10 times on the training set, and the best performing model on the validation set was used for testing and model comparison. To ensure that this training scheme is appropriate, we have also implemented two other strategies from the literature: early stopping, and the strategy used in Torada *et al.* for ImaGene [38], which they refer to as simulation on-the-fly, and evaluated model accuracy (Figs Q and R in S1 Text). We see that our strategy performs on par with early stopping, and on average performs better than the ImaGene strategy. To test the performance of each model, accuracy, area under the ROC curve, and correlation estimates were calculated using the balanced test set. 95% confidence intervals for accuracy were computed using $\hat{p} \pm 1.96 \times \sqrt{\frac{\hat{p}(1-\hat{p})}{N_t}}$ where $\hat{p}$ is the computed accuracy and $N_t$ is the size of the test set. $N_t = 10,000$ for the single population demographic model while $N_t = 2,000$ for the three population model. All models were trained on a combination of Nvidia RTX 2060 and 3090 GPUs.

## Implementation of summary statistics

Each summary statistic was implemented and applied to the raw haplotype data using the `scikit-allel` python package [60]. iHS values were standardized based on neutral simulations for the demographic model of interest, learning the mean and variance for standardization for each derived allele count from 0 to 128. To produce a single value for the whole genome region, we took the maximum absolute value of all standardized iHS values across the region.

For each summary statistic, a decision threshold was used to classify a region as sweep or neutral based on the computed statistic. This threshold was found by numerically computing the optimal decision threshold which maximized the accuracy of the method on the training set, and performance was evaluated on the test set.

## Methods of interpretation

Where possible, we plotted visualizations of the convolution kernel and dense weights. In the case of more complex models with large parameter counts, we used the SHAP DeepExplainer [42] to generate visual explanations for model predictions. For a given deep learning model and input, SHAP values are used to measure the importance of input features to the final prediction. When visualized, the red pixels may be interpreted as feature locations contributing to an increase in the model's output (towards sweep classification) while blue pixels contribute to a decrease (towards neutral classification). For each simulation type, we provided 20 neural and 20 sweep images for use by DeepExplainer as background samples.

## Supporting information

**S1 Table. Convolutional neural networks for detection of selective sweeps.**
(XLSX)

**S1 Text. Supplementary material.** Fig A. Comparison of Imagene with mini-CNN. Table A. Model accuracy for decreasing levels of complexity, for single-population demographic model with selection coefficient $s = 0.01$. Fig B. SHAP explanations for Imagene predictions without row-sorting. Table B. Model accuracy for decreasing levels of complexity, for single-population

demographic model with selection coefficient $s = 0.01$, trained without row-sorting. Fig C. Performance correlation for summary statistics and CNN approaches under image resizing. Fig D. Visualization of model performances across all demographic models for selection coefficient of 0.01. Fig E. Visualization of model performances across all demographic models for selection coefficient of 0.005. Fig F. Simulations of CEU and YRI under the three-population demographic model match the site frequency spectrum of 1000 Genomes populations. Fig G. SHAP explanations for Imagene predictions under image resizing. Fig H. Visualization of mini-CNN dense layer under image resizing. Fig I. Visualization of mini-CNN dense layer under zero-padding. Fig J. SHAP explanations for Imagene predictions under zero-padding. Fig K. Performance correlation for summary statistics and CNN methods under zero-padding. Fig L. SHAP explanations for Imagene predictions with trimming for standardizing image width. Fig M. Full images corresponding to the cropped images in Fig 4. Fig N. SHAP visualizations for Imagene trained to distinguish between hard/soft sweeps and neutrality. Fig O. Visualizations of mini-CNN dense layer, comparing Hard vs Neutral and Hard vs Soft sweep classification. Fig P. Sample Complexity Analysis. Fig Q. Visualization of model performances across all demographic models and training strategies, for selection coefficient of 0.01 Fig R. Visualization of model performances across all demographic models and training strategies, for selection coefficient of 0.005.
(PDF)

## Acknowledgments

We are grateful to Sara Mathieson, Arthur Sugden, and members of the Ramachandran lab for valuable discussion and feedback. We are also grateful to Matteo Fumagalli for sharing resources related to Imagene.

## Author Contributions

**Conceptualization:** Ryan M. Cecil, Lauren A. Sugden.

**Data curation:** Ryan M. Cecil.

**Formal analysis:** Ryan M. Cecil.

**Investigation:** Ryan M. Cecil.

**Methodology:** Ryan M. Cecil, Lauren A. Sugden.

**Software:** Ryan M. Cecil.

**Supervision:** Lauren A. Sugden.

**Writing – original draft:** Ryan M. Cecil, Lauren A. Sugden.

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
