## [Decision Letter · Decision Letter 0]

16 Apr 2023

Dear Mr. Cecil,

Thank you very much for submitting your manuscript "On convolutional neural networks for selection inference: revealing the lurking role of preprocessing, and the surprising effectiveness of summary statistics" for consideration at PLOS Computational Biology.

As with all papers reviewed by the journal, your manuscript was reviewed by members of the editorial board and by several independent reviewers. In light of the reviews (below this email), we would like to invite the resubmission of a significantly-revised version that takes into account the reviewers' comments.

We cannot make any decision about publication until we have seen the revised manuscript and your response to the reviewers' comments. Your revised manuscript is also likely to be sent to reviewers for further evaluation.

Sincerely,

Piero Fariselli

Academic Editor

PLOS Computational Biology

James O'Dwyer

Section Editor

PLOS Computational Biology

Reviewer's Responses to Questions

**Comments to the Authors:**

Reviewer #1: The review is uploaded as an attachment.

Reviewer #2: The manuscript explores the inner working of CNN for the detection of selective sweeps. The exposition is clear and complete and I find the insights original enough. I have a just some minor suggestions/requests:

1) There is a big gap in performance between the two selected selection coefficients, with the 0.005 coefficient being very hard to detect for all methods. Did you try to use more haplotypes as input to see if they can increase the detection power of some methods? I think that would be an interesting addition to your work.

2) Please spruce up a bit the github repository: provide a readme with a couple of notes on how to run the code, add a requirement file or at least a list of needed modules, and maybe remove or complete the empty script (reproduce.sh) at the top level.

3) If possible try to get the figures and tables closer to the text where they are mentioned, it would help the readers!

Reviewer #3: This paper by Cecil and Sugden is concerned with the accuracy and interpretability of deep neural networks (DNN) for population genetic inference. Here, the authors examine the performance of a fairly similar convolutional neural network (CNN), a type of DNN that has been applied to various population genetic tasks in recent years. The authors find that, on their test problem, the CNN performs no better than a simple summary statistic, and also point out ways in which the input representation affects performance.

I enjoyed this paper's emphasis on the importance of interpretability, and their discussion of using CNNs as more than just black boxes, an area in which our field is clearly lagging behind. However, I have some concerns about the generality of the results. Indeed, a close examination of the authors' claims and how they relate to what has previously been shown in the literature strongly suggests that these results do not generalize across CNNs. Nor should we expect these results to generalize across problems, as the authors are looking only at very recent positive selection. I do appreciate the authors' concern about artifacts in deep learning, as this has been well documented in other contexts, and is an issue we should be concerned with in population genetics. But as I argue below, this manuscript contains no analyses that touch upon this subject, instead containing a very misleading argument about the number of segregating sites within a genomic window.

I think that there are important arguments to be made about the tradeoff between interpretability and statistical power, and also it is almost certainly the case that for some tasks more interpretable methods will perform as well as black box approaches (i.e. those problems for which we have a large number of statistics available, and there is a statistic or combination thereof that is likely to perform well in the parameter range of interest, as is certainly the case in sweep detection). Unfortunately, this paper does not make any such arguments very effectively, and offers little in the way of convincing data.

Because I feel that this paper has a number of fundamental flaws, I have focused my review only on major concerns, which I detail below:

1) First, the author's description of the history of sweep detection is misleading, painting a picture that the only methods were summary stats based on the SFS, prior to the advent of the EHH statistic, inflating the importance of the latter. There is no mention of LD-based tests (e.g. https://pubmed.ncbi.nlm.nih.gov/9215920/), and especially Hudson's haplotype test (https://pubmed.ncbi.nlm.nih.gov/8013910/), based on shared haplotypes. There were SFS-based and non-SFS-based statistics devised both before and after the advent of EHH, with may of the latter having nothing to do with EHH. I think this is important here because we want to accurately convey that there have been a vast array of approaches for sweep detection that have attacked the problem from many different angles, and this is relevant to my point above about sweep detecting being a problem that already has a large number of interpretable features that one could use without relying on a deep neural network.

2) The authors may well have a CNN that performs no better than Garud's H1 on their task of detecting sweeps, but one cannot generalize based on this observation in the way that the authors are clearly intending. Indeed, there is a clear demonstration that some CNNs perform VASTLY better than H12. For example, the Flagel et al. CNN performs at least as well as a S/HIC, a method based on a vector of summary statistics and which in turn shows a huge increase in accuracy relative to H12 (e.g. Figures 2 and 3 from https://journals.plos.org/plosgenetics/article?id=10.1371/journal.pgen.1005928). (I know that H12 is not the same as H1, but they are so similar and the difference in power between the two is pretty negligible, so that does not matter here.)

3) Related to point 2 above: the fact that at least some CNNs are much more powerful than Garud's H1, means that the conclusion the authors have reached that "in the process of row-sorting, the CNN is being set up to mimic the behavior of Garud’s H1" clearly cannot be true in general, even for CNNs that sort rows by similarity. It is true that not all CNNs have sorted in the same manner is Imagene, but again, so that may matter somewhat, but the point is that the authors are overgeneralizing and drawing conclusions that are clearly already contradicted in the literature.

4) One's chosen summary stat(s) may not be appropriate given the problem at hand. For example, in some parameterizations (e.g. older sweeps), Garud's H1 will have very little power. If you know exactly which statistic or combination thereof will be most powerful in the relevant region of the parameter space, then this is not much of a problem, as you can choose your statistic(s) appropriately. But this may not always be the case in practice...

5) And what if we don't even have an adequate set of summary statistics for the task at hand? Consider the introgression problem from Flagel et al, where a CNN performs far better than a method using a large vector of summary statistics. The authors pay some lip-service to this possibility, but in a fairly dismissive tone. This is unfortunate, because I think this is probably the real benefit of CNNs: even if for your problem/parameterization we have not already discovered an adequate set of statistics, we have a way forward. Perhaps the authors are trying to argue that we should instead spend more time developing such statistics so we can engineer better feature vectors, and while I would be sympathetic to this argument, I do not find that the analyses in this paper would offer convincing support for it.

6) Finally, I am somewhat taken aback by the notion that encoding the number of segregating sites in a physical region is somehow artifactual. The density of polymorphism is of course informative about natural selection, and the authors seem to be implying that the researchers who chose to design CNNs that very clearly represent this information were somehow accidentally "artificially inflating" performance, a notion that strains credulity. It is quite trivial to point out that this padding approach essentially includes the number of segregating sites as an input feature. Indeed, some network architectures have a separate input channel that in some way encodes the position of segregating sites, and when that is padded the number of segregating sites is even more readily extracted by the network. Do the authors truly believe that in these cases the designers of those networks were unaware that this would be happening. This is obviously not the case, and the number of segregating sites is instead intentionally being given to the network because it is useful for answering the question at hand, as regions affected by positive selection will have relatively few polymorphisms. Of course, there may be some situations where one does not want to use this information, and the authors could make an argument about this if they wish. But, to me, the blanket implication that a network’s encoding of the number of polymorphisms in a window is somehow undesirable or unintended is almost akin to stating that Watterson's estimator contains entirely artifactual information. In short, the authors’ unnecessarily provocative claim about so-called artifactual information is confusing, distracting, and completely unsupported, and I strongly urge the authors to remove it.

In short, despite stating early on in the Introduction that "the choice of CNN architecture varies widely across these works" and that this in part "makes direct comparison between the approaches tricky", the authors have used a very small number of analyses on a single CNN to make a straw-man argument, which they then overgeneralize from to imply that sets of statistics, or even single statistics, have more power than neural networks (which may be the case for some especially easy parameterizations of especially well-studied problems), even though their results and conclusions clearly do not hold water when examined in the context of previously published results in this area.

**Have the authors made all data and (if applicable) computational code underlying the findings in their manuscript fully available?**

Reviewer #1: Yes

Reviewer #2: Yes

Reviewer #3: None

PLOS authors have the option to publish the peer review history of their article (what does this mean?). If published, this will include your full peer review and any attached files.

Reviewer #1: No

Reviewer #2: No

Reviewer #3: No
---

## [Decision Letter · Decision Letter 1]

13 Aug 2023

Dear Mr. Cecil,

Thank you very much for submitting your manuscript "On convolutional neural networks for selection inference: revealing the lurking role of preprocessing, and the surprising effectiveness of summary statistics" for consideration at PLOS Computational Biology.

As with all papers reviewed by the journal, your manuscript was reviewed by members of the editorial board and by several independent reviewers. In light of the reviews (below this email), we would like to invite the resubmission of a significantly-revised version that takes into account the reviewers' comments.

We cannot make any decision about publication until we have seen the revised manuscript and your response to the reviewers' comments. Your revised manuscript is also likely to be sent to reviewers for further evaluation.

Sincerely,

Piero Fariselli

Academic Editor

PLOS Computational Biology

James O'Dwyer

Section Editor

PLOS Computational Biology

Reviewer's Responses to Questions

**Comments to the Authors:**

Reviewer #1: Review is uploaded as an attachment.

Reviewer #3: In their revision, Cecil and Sugden have added several helpful analysis to their paper, "On convolutional neural networks for selection inference: revealing the lurking role of preprocessing, and the surprising effectiveness of summary statistics." More importantly, the authors state in their response that they have toned down their claims, which I appreciate. However, upon carefully examining the actual changes made in the manuscript, I found that there were key points in the manuscript that are still misleading:

1) Saying that they cannot conclude that their findings would generalize to other problems (or more challenging formulations of the sweep detection problem) is not sufficient, as it leaves the door open to the possibility that their findings might generalize, when it is in fact quite clear that they would not.

Although the authors pay lip service to the fact that there are more statistics for detecting sweeps than for some other problems (e.g. detecting introgressed loci), a more complete discussion of what is going on in these other problems is vitally important as this would better characterize the real motivation for using deep learning in population genetics: not only is there not a single statistic that can recapitulate the performance of a DNN for detecting introgression, even a combination of such statistics does not can get anywhere close. Until such statistics have been designed, deep learning will be a more powerful alternative. In other words, the motivation is that one need not wait for the ideal statistic or set thereof to be derived, but if and when those statistics are discovered, more interpretable methods with similar performance may be possible (and preferable).

2) I was surprised to see that the authors were still insinuating that neural networks that contain information about the density of polymorphism are in fact using unintended/artifactual information, when this is clearly not the case. Researchers making the design choices to include padding/positional information know that their input contains the density of segregating sites. The authors can state that they wish to inform those who may be new to neural network design about this, which would be useful, but manuscript still seems to be implying that previous studies were unintentionally using this information, and this needs to be corrected. Indeed, the title's mention of the "lurking role" of preprocessing contributes to this unnecessarily accusatory tone and I find it to be unjustified and inappropriate.

On a related note, It is helpful that the authors point out the importance of training data with variable mutation rates, but make no effort to mention previous studies that have done this (e.g. for the sweep-detection problem in Flagel et al., theta varied across an order of magnitude).

3) Yes, when developing a new method, design decisions matter, and good designers will make those intentionally and carefully. The authors' premise seems to be that researchers using deep learning are putting no thought into their designs, and simply outsourcing all of their decision making to the training process. I strongly disagree with this view. For example, deep learning gives designers far greater flexibility in designing input representations. In the abstract, the authors say, "We conclude that human decisions still wield significant influence on these methods," which is of course obvious and well known. It would be more accurate to say that your study examines some of the ways that some of these decisions can influence performance.

4) I also want to point out that the authors' arguments about window size and H12 are unconvincing, given that Garud et al. (2015) examined H12 in 400-SNP windows.

**Have the authors made all data and (if applicable) computational code underlying the findings in their manuscript fully available?**

Reviewer #1: Yes

Reviewer #3: None

PLOS authors have the option to publish the peer review history of their article (what does this mean?). If published, this will include your full peer review and any attached files.

Reviewer #1: No

Reviewer #3: No
---

## [Decision Letter · Decision Letter 2]

26 Oct 2023

Dear Mr. Cecil,

We are pleased to inform you that your manuscript 'On convolutional neural networks for selection inference: revealing the effect of preprocessing on model learning and the capacity to discover novel patterns' has been provisionally accepted for publication in PLOS Computational Biology.

Best regards,

Piero Fariselli

Academic Editor

PLOS Computational Biology

James O'Dwyer

Section Editor

PLOS Computational Biology

Reviewer's Responses to Questions

**Comments to the Authors:**

Reviewer #1: The authors have adequately addressed my remaining comments in the latest round of revision. The title now reflects the key takeaways from this study quite well.

One last minor point regarding Fig. 1 panel c, figs S4&S5, I understand that best plotting practices might advise against axis truncation, but I still believe that adding a "zoomed-in" view of the plots (perhaps a panel chart from the link to Duke Library the authors referenced) can greatly improve visualization (particularly for Fig. 1C, since it's also a main figure).

Reviewer #3: The authors have addressed all of my comments to my satisfaction. I find that this revision now paints a more balanced picture of an important problem.

I am still somewhat confused by 10 kb window claim, as a few simple simulations demonstrate that the footprint of a sweep with the parameters given in the Garud et al. example will extend for at least 100 kb, if not more. It is possible that Garud et al. mistakenly used centi-Morgans instead of Morgans in this equation? But I don't think this is an important issue for the present submission.

**Have the authors made all data and (if applicable) computational code underlying the findings in their manuscript fully available?**

Reviewer #1: Yes

Reviewer #3: None

PLOS authors have the option to publish the peer review history of their article (what does this mean?). If published, this will include your full peer review and any attached files.

Reviewer #1: No

Reviewer #3: No

---

## [Editor Report · Acceptance letter]

21 Nov 2023

PCOMPBIOL-D-23-00312R2 

On convolutional neural networks for selection inference: revealing the effect of preprocessing on model learning and the capacity to discover novel patterns

Dear Dr Cecil,

I am pleased to inform you that your manuscript has been formally accepted for publication in PLOS Computational Biology. Your manuscript is now with our production department and you will be notified of the publication date in due course.

With kind regards,

Zsofi Zombor
